# Enhancement of Diversity in Production and Application Utilizing Electrolytically Polymerized Rubber Sensors with MCF: 1st Report on Consummate Fabrication Combining Varied Kinds of Constituents with Porous Permeant Stocking-Like Rubber

**DOI:** 10.3390/s20174658

**Published:** 2020-08-19

**Authors:** Kunio Shimada, Ryo Ikeda, Hiroshige Kikura, Hideharu Takahashi

**Affiliations:** 1Faculty of Symbiotic Systems Sciences, Fukushima University, 1 Kanayagawa, Fukushima 960-1296, Japan; 2Institute of Innovative Research, Tokyo Institute of Technology, 2-12-1 Ookayama, Meguro-ku, Tokyo 152-8550, Japan; ikeda.r.ah@m.titech.ac.jp (R.I.); kikura@lane.iir.titech.ac.jp (H.K.); htakahashi@lane.iir.titech.ac.jp (H.T.)

**Keywords:** sensor, diene rubber, surfactant, porous, permeation, electrolytic polymerization, magnetic compound fluid (MCF), hybrid skin (H-skin)

## Abstract

To satisfy the requirement of haptic sensibility in rubber such as in the proposed hybrid skin (H-Skin), the authors have demonstrated a new method for solidifying rubber using electrolytic polymerization together with configured magnetic clusters of magnetic compound fluid (MCF) incorporated into the rubber by the application of a magnetic field. However, the rubber and magnetic fluid (MF) involved in the MCF rubber were water-soluble. In addition, the authors have demonstrated the practicability of using electrolytic polymerization with an emulsifier, polyvinyl alcohol (PVA), in which natural rubber (NR) or chloroprene rubber (CR) and silicone rubber (Q) can be mixed as water-soluble and water-insoluble rubbers, respectively. In this study, to enhance production, the feasibility of solidifying rubber by electrolytic polymerization is verified using varied water-insoluble rubber, varied water-insoluble MF, and varied surfactants to aid emulsion polymerization, except in the case of other kinds of rubber and MF which have been demonstrated until recent by the authors. Based on these diverse constituents, the authors propose a consummate fabrication process for multi-layered MCF rubber, which involves porous stocking-like rubber that can be permeated by any liquid. The investigation of this application is presented in the sequential second report.

## 1. Introduction

Sensors require haptic sensibility in rubber under normal or shear motions, including sensibilities to temperature or photoreaction. The haptic sensibility that is provided to robots predominantly corresponds with sensing normal or shear forces and temperature [1,2,3]. These sensibilities are related to the five types of touch sensations in the human skin: tactile, baresthesia, algometry, warm, and cold [4]. When sensibility is incorporated into the skin of a robot, the material that is feasible to the outer artificial skin installed on it is termed sensitive skin, smart skin, or electronic skin (E-Skin) [5]. As an alternative artificial skin, the authors have also proposed a hybrid skin (H-Skin) with multiple functionalities, including flexibility, thermo-sensitivity, photovoltaics, and ability to sense modalities to its forces [6,7,8].

H-skin is produced by a new method using solidified rubber by electrolytic polymerization, together with the configuration of magnetic clusters of metal particles integrated into the rubber by the application of a magnetic field [9,10,11]. The resulting material is named magnetic compound fluid (MCF) rubber. This novel solidification method differs from the conventional vulcanization technique, where sulfur is used on the field for the normal production of solid rubber [12]. However, in the case of MCF, it possesses nm-ordered magnetite (Fe_3_O_4_) particles that are obtained using a magnetic fluid (MF) during compounding, and μm-ordered metal particles, such as Fe and Ni. Because it is an effective magnetic-responsive intelligent fluid, MCF has been proposed as an alternative in many engineering applications, such as dampers and polishing machines [13]. By applying a magnetic field on the MCF rubber during electrolytic polymerization, the magnetic particles of MCF are fabricated as heterostructures like in many needle-like clusters so that the electric and photovoltaic properties of MCF rubber are enhanced, as well as its mechanical and electrical properties becoming anisotropic. Furthermore, to fabricate an MCF rubber sensor, a novel adhesion technique between rubber and metal has been proposed with hydrate-containing fabricated metallic or non-metallic elements under electrolytic polymerization to adhere plate- or wire-type electrodes to the rubber without the occurrence of detachment caused by large tension [14]. On one hand, the rubber that has been provided with electrolytic polymerization is water-soluble, which includes natural rubber (NR) and chloroprene rubber (CR) categorized as diene rubber. On the other hand, the water-insoluble rubber, silicone rubber (Q), can be electrolytically polymerized to be solidified together with water-soluble rubber such as NR or CR by mixing polyvinyl alcohol (PVA) as a surfactant or an emulsifier [15].

However, the authors have dealt with the combination of Q, NR, CR, PVA, and water-base MF alone until recent [15]. Other diverse combinations must be managed for example, the combination of water-insoluble rubber and water-insoluble MF, water-soluble rubber and water-insoluble liquid, that of other emulsifiers, and so on, which are not elucidated yet. It is critical to validate the practicability of electrolytic polymerization for the solidification of rubber with various combinations because the production of solidified rubber is convenient, whichever diverse kinds of rubber and MF are used. From point of view of other researches to have been conducted on so far, it is significant to develop the emergence of rubber blends by mixing two or more than polymers together in the field of polymer and rubber sciences so that many investigations have been conducted on, for example, by combining water-soluble and insoluble rubbers [16]. Carbon black or nanotube as well as ethylene-propylene-diene rubber (EPDM) has been often used, and the former was for its reinforcement of the rubber [17]. The method how to solidify these rubbers was the vulcanization with sulfur. As for the combination of MF and rubber, Q has been often used [18,19]. On these investigations, the problem of the combination of rubber and MF as well as the combination of water-soluble and insoluble rubbers, has not been focused on vulcanization. Their primary purposes were to advance highly the functional properties, mechanical or electrical properties. Therefore, it is meaningful to verify and construct the mechanism of the combination of the diverse kinds of rubber and MF in our present study. In addition, little attention to the investigation of the combination of varied emulsifier with the rubber has been given to the point. PVA has the role played by the emulsion polymerization so that PVA is significant in the electrolytic polymerization of combined diene and non-diene rubbers. Conventionally has MF the surfactant of oleic acid that has also the role played by the emulsion polymerization. It remains an unsettled question how the emulsifier or surfactant combines with the rubber. Moreover, it is important to address the fabrication of the MCF rubber sensor with various combinations of rubber, MF and emulsifier. Because the fabrication of the MCF rubber sensor by the combination of Q, NR, CR, PVA, and water-base MF alone has been proposed [14].

Regarding the fabrication, if the MCF rubber initially involves a liquid-type dopant, the MCF rubber sensor can have a variety of properties, which are mainly divided into three types by the kinds of dopants: conductive, piezo, and battery types. The solid type dopant only needs to be mixed; however, the feasibility of involving the liquid-type dopant in the MCF rubber might be realized by the permeation of liquids. In order to do that, we might utilize the technique for making rubber porous. From point of view of other studies to have been conducted on so far, various ideas have been proposed. For example, NaCl particles that are initially mixed in Q and vulcanized are washed and removed by water to obtain porosity [20]. A porous structure is also formed with composites blended by rubber and cement, which are mixed with asphalt and rubber for concrete pavement; however, this rubber is much stiffer than the proposed MCF rubber [21]. Additionally, a porous material can be fabricated by mixing ammonia stabilizer in NR with a carbon nanotube fiber and by the evaporation of the stabilizer when heat is applied [22]. The porous material containing cellulose fiber is mixed with styrene-butadiene rubber (SBR) and asphalt, and porosity can be obtained by evaporating water in the mixture [23]. Porous rubber has been predominantly used for the permeability of gas and liquid or composites with concrete or asphalt. However, our proposing idea in the present study of electrolytically polymerized porous and permeable rubber has not yet been proposed. We must address the fabrication of the MCF rubber sensor with utilizing the porous MCF rubber involving a liquid-type dopant. The production of the porous rubber and the permeation of dopants into the rubber are other typical subjects in the present study.

In this study, the authors investigate the viability of: (1) electrolytic polymerization technique for the solidification of diverse rubbers with diverse combinations of various rubbers and MFs using various emulsifiers; (2) the fabrication process of the MCF rubber sensor with their combinations and the MCF rubber permeated by liquids, which are the novelty of the current work. In addition, as the technique and process are enhanced for diverse applications, the authors try to apply the fabricated MCF rubber sensor to fields related to normal or shear forces and temperature, which is presented in the second report [24].

## 2. Electrolytic Polymerization

### 2.1. Rubber

Rubber is divided into diene and non-diene rubbers or natural and synthetic rubbers and categorized as shown in Figure A1 (Appendix A). For diene rubber, the latex in a liquid state as in NR, CR, isoprene rubber (IR), and butadiene rubber (BR) can be solidified by electrolytic polymerization. This can be achieved because diene rubber contains C=C bonds and water, which has been elucidated in the previous research [9,10,11,12]. These latexes are electrolytes that contain enough water to be provided by electrolysis. Nitrile rubber (NBR) and SBR, however, cannot be provided with electrolytic polymerization and cannot be solidified because they have potentially high viscosity. In contrast, non-diene rubber contains C-C bonds such that electrolytic polymerization cannot be utilized. However, by utilizing PVA and mixing water-soluble diene-rubber such as NR-latex or CR-latex, the silicone oil in a liquid state (Q) can be solidified by electrolytic polymerization. Basically, Q is structured as the basis of dimethylpolysiloxane (PDMS) and PVA is an emulsifier, allowing PDMS and PVA to be combined by emulsion polymerization. The anionized hydroxy group of PVA and NR-latex or CR-latex are bonded by hydrogen boding. This mechanism has been elucidated in previous research [15]. However, when the water-soluble diene-rubber rubber is not mixed with water-insoluble non-diene rubber, electrolytic polymerization might become possible, provided the non-diene rubber contains a fair portion of C=C bonds in its molecular structure and liquid state. As another way of validating electrolytic polymerization using PVA and mixing NR-latex or CR-latex except for Q, the electrolytic polymerization of urethane rubber (U) is demonstrated in this study.

The U used in this study was polyurethane (C7 with hardness measured by C-type durometer, hardness meter (Kobunshi Keiki, Co., Ltd., Kyoto, Japan), Exseal Co. Ltd., Gifu, Japan). It was emulsified by PVA, with either NR-latex, CR-latex, or their composite latexes bonded with the anionized hydroxy group of the PVA by hydrogen boding. The MCF rubber liquid consisted of 3 g Ni powder with particles in the order of microns and bumps on the surface (No. 123 by Yamaishi Co. Ltd., Noda, Japan), 3 g PVA, 0.75 g water-based MF with 40 wt% Fe_3_O_4_ (W-40, Ichinen-Chemicals Co., Ltd., Shibaura, Japan), 3 g CR-latex (671A, Showa Denko Co. Ltd., Tokyo, Japan), 0.5 g TiO_2_ (Anatase type, Fujifilm Wako Pure Chemical Co., Ltd., Osaka, Japan), 3 g NR-latex (without sulfur, Ulacol, Rejitex Co. Ltd., Atsugi, Japan), or S-500 (NR-latex with sulfur, Rejitex Co. Ltd., Atsugi, Japan). U and PVA must be mixed in an advance combination as in Q [15] because if U is combined with a single NR-latex or CR-latex, or their mixture, they are solidified during stirring. They must also be stirred under cooling conditions with ice water. Subsequently, the MCF rubber liquid was electrolytically polymerized, where a static magnetic field of 312 mT was applied to a pair of stainless electrodes with a 1 mm gap using permanent magnets as paired opposites via the application of a constant electric field at 20 V and 2.7 A, for 5 min.

Incidentally, as provided throughout the experiment of this study, the magnetic field strength and electric current can be determined by optimum magnitude, as has been presented in the previous studies [15]. It can be measured with the Gauss meter probe, which is a traditional instrument for magnetic field measurement. Based on many experiments, 312 mT was confirmed to be the optimal magnetic field strength in the production method proposed in this study. With a gap of 1 mm between electrodes, permanent magnets were paired opposites via a constant electric field at 2.7 A with a voltage range of 6–30 V, for 5–30 min. Therefore, 312 mT and 2.7 A were used during this study.

Figure 1 shows images of liquid MCF rubber before electrolytic polymerization and electrolytically polymerized MCF rubber. As shown in Figure 1a, the combination of diene and non-diene rubber latexes with MCF results in a highly uniform dispersion. As seen in Figure 1f, the rubber molecules exhibit mutual conjuncts to be crosslinked.

The electrical property of the voltage induced inner MCF rubber at compression was measured using the same NFE experimental apparatus as in the previous study [9]. NFE specified that the upper electrode is moved to touch the MCF rubber sensor, which is squeezed between the electrodes, onto the lower one by an actuator at a pressing speed of 10 mm/min. The normal pressing force was measured by a load cell, which was installed in the actuator. The actuator utilizes the commercial, small-size tensile testing machine (SL-6002, IMADA-SS Co. Ltd., Toyohasi, Japan). The voltage between the electrodes can be measured without installing a power supply. The paired electrodes used had the same 7 mm × 7 mm square form. The measurement was for induced voltage related to piezo-electricity, which was evaluated as built-in electricity, built-in voltage, and current [25]. The built-in electricity occurred by ionized molecules, particles of rubber latex and oleic acid-coated around Fe_3_O_4_ of MF, water, and PVA. They play the role of an acceptor-like p-type semiconductor (corresponding to A^−^ as presented in the previous study [6,12]), or a donor-like n-type semiconductor (corresponding to D^+^ [6,12]). Figure 2 shows that the induced voltage applies pressure repeatedly. The resulting MCF rubber just after production contains induced voltage which is independent of pressure, although, after some lapses, become a piezo-element that depends on pressure. At any rate, the electrolytic polymerization technique that utilizes PVA and mixing water-soluble diene rubber such as NR-latex or CR-latex with non-diene rubber is feasible.

### 2.2. Water-Insoluble Liquid and MF

The roles that NR-latex or CR-latex and Q or U play can be combined through the medium of emulsion polymerization with a PVA-enabled non-conductive rubber such as non-diene rubber to be solidified by the application of an electric field. This role connotes the possibility of mixing water-insoluble liquid and water as shown in Figure 3a,c,e, and the practicability of mixing water-insoluble MF and water-base MF as shown in Figure 4a,c. Various kinds of MFs exist, and in this study, the authors deal with a well-known base MF and its solvent as a water-insoluble liquid. As shown in Figure 3, the liquid has 3 g PVA, 0.75 g water, 0.75 g Fe_3_O_4_ powder (Fujifilm Wako Pure Chemical Co., Ltd., Osaka, Japan), 3 g Ni, and 0.75 g water-insoluble liquid (kerosene, alkyl naphthalene, paraffin). As represented in Figure 4, the liquid has 3 g PVA, 0.75 g MF (W40), 3 g Ni, and 0.75 g water-insoluble MF (kerosene base MF (MSGS60, Ferrotec Co., Ltd., Tokyo, Japan), and alkyl naphthalene base MF (A500, Sigma Hi-Chemical Co., Ltd., Kanagawa, Japan)). The mixing process is presented in Figure 5. Consequently, via an emulsion polymerization medium by PVA, water-insoluble liquid or water-insoluble MF can be combined with an inner diene type MCF rubber, such as a water-soluble type MCF rubber. In this mechanism, the water-insoluble liquid and PVA can be combined by the emulsion polymerization of PVA, and the anionized hydroxy group of PVA and water are bonded by hydrogen boding. However, regarding Figure 3b,d,f and Figure 4b,d, when a magnetic field is applied, the liquid seems to be divided into magnetic and non-magnetic materials because of the mixed magnetic particles and non-magnetic materials. Under these phenomena, it is strange that the MCF rubber containing water-insoluble liquid or MF possessed electrical properties and was electrolytically polymerized. This anomaly is indicated in sequential figures.

In the case without the application of a magnetic field, via an emulsion polymerization medium by PVA, water-insoluble liquid or water-insoluble MF can be combined with an inner diene type MCF rubber, such as a water-soluble type MCF rubber. As regards the former, the combined MCF rubber liquid with or without a magnetic field and the electrolytically polymerized MCF rubber, are shown in Figure 6. In the case of the latter, it is represented in Figure 7 while the method of mixing is shown in Figure 8. In Figure 6, the MCF rubber liquid consisted of 3 g Ni, 3 g PVA, 3 g 671A, 3 g NR-latex, 0.75 g Fe_3_O_4_, and 0.75 g water-insoluble liquid (kerosene, alkyl naphthalene, and paraffin). In Figure 7, the MCF rubber liquid consisted of 3 g Ni, 3 g PVA, 3 g 671A, 3 g NR-latex, 0.75 g water-insoluble MF (MSGS60, A500, another kerosene base MF (HC50, Ichinen-Chemicals Co., Ltd.), and diesters base MF (DS50, Sigma Hi-Chemical Co., Ltd.). With a 1 mm gap between the electrodes, a 312 mT magnetic field, and an electric current field at 20 V and 2.7 A were applied for 5 min. In this mechanism, water-insoluble liquid (or MF) and PVA can be combined by the emulsion polymerization of PVA, and the anionized hydroxy group of PVA and NR-latex (or CR-latex) are bonded by hydrogen boding.

When a magnetic field is applied before electrolytic polymerization, the MCF rubber liquids, as shown in Figure 6b,h,n and Figure 7b,h,n,t, are less clearly divided than those in the cases without using water-soluble rubber, as shown in Figure 3b,d,f and Figure 4b,d. In contrast, when electric polymerization is applied under a magnetic field, the MCF rubber is not distinctively divided into two, areas such as magnetic and non-magnetic materials.

As for the MCF rubbers of Figure 6 and Figure 7, the pressure was repeatedly applied by the induced voltage, as shown in Figure 9a and Figure 10a. Aside from the induced voltage causing piezo-electricity, there was another effect on a piezo-element, piezo-resistivity [25]. A piezo-resistive element requires the application of voltage by a power supply, thus producing the changes in resistivity experienced by the piezo-element. This element is essentially an electric conductor that allows electrons to percolate through the material, and the mechanism of this electric conductivity is explained mainly by percolation or tunnel theory [6]. By applying the voltage to the MCF rubber, the authors measured the electrical current passing through the MCF rubber at compression using the same NFE experimental apparatus as in the measurement of piezo-electricity, as shown in Figure 9b and Figure 10b. The MCF rubber induced voltage and electric current, which are dependent on pressure, and the response to pressure is different depending on the kind of water-insoluble liquid and MF. In any case, the diene type MCF rubber that is mixed by water-insoluble liquid or water-insoluble MF, which uses the emulsion polymerization of PVA, has a feasibility of haptic sensing.

Next, the authors investigated the possibility of haptic sensing in the case of non-diene type MCF rubber, such as water-insoluble type MCF rubber, which is mixed with a water-insoluble material (MF and liquid) through the medium of emulsion polymerization by PVA and by mixing water-soluble rubber, such as NR-latex or CR-latex. When the emulsion polymerization of PVA is used to mix MCF rubber with Q and NR-latex or CR-latex, the resulting combination of water-insoluble MF is investigated. In this mechanism, water-insoluble MF and PVA can be combined by the emulsion polymerization of PVA, and the anionized hydroxy group of PVA and NR-latex (or its mixture with CR-latex) are bonded by hydrogen boding. KF96 is processed as Q, which is a pure silicone oil without any silane, such that it cannot be solidified by mixing it with a curing agent. As for Figure 11, the MCF rubber liquid consisted of 3 g Ni, 3 g PVA, 3 g NR-latex, 3 g KF96 (1000 cSt, Shin-Etsu Chemical Co. Ltd., Tokyo, Japan), and 0.75 g water-insoluble MF (NSGS60, A500, HC50, DS50). In the case represented in Figure 12, the MCF rubber liquid consisted of 3 g Ni, 3 g PVA, 3 g 671A, 3 g NR-latex, 3 g KF96 (1000 cSt), 0.75 g MF (W40), and 0.75 g water-insoluble MF (MSGS60, A500). The method of mixing is presented in Figure 13. A 312 mT magnetic field and an electric field at 20 V and 2.7 A were applied for 5 min with a 1 mm gap between the electrodes. When Q is combined with MCF rubber liquids by applying a magnetic field before electrolytic polymerization, as shown in Figure 11b,h,n,t and Figure 12b,h, the resulting combination becomes more uniform than those in the cases without Q such as those in Figure 6b,h,n and Figure 7b,h,n,t. Although the surfaces of electrolytically polymerized rubber facing anode seen in Figure 11e,k,q,w and Figure 12e,k are smooth, those facing cathode seen in Figure 11f,l,r,x and Figure 12f,l are rougher than those in the cases without Q such as those in Figure 6f,l,r and Figure 7f,l,r,x. In the case involving the combination of diene and non-diene rubbers, water-insoluble MF or the compound of water-insoluble and water-based MFs can be mixed so that the MCF rubber can be electrolytically polymerized.

As for the MCF rubbers shown in Figure 11 and Figure 12, the induced voltage and electric current as in piezo-electricity and piezo-resistivity, respectively, which repeatedly apply pressure, are shown in Figure 14 and Figure 15. The change in induced voltage due to pressure is small, whereas the electric current has an unexpected surge response to pressure. At any rate, non-diene type MCF rubber, combined with water-insoluble or water-based MF by the emulsion polymerization of PVA and by mixing soluble rubber such as NR-latex or CR-latex, can have the feasibility of sensing.

### 2.3. Surfactant

From the above results, the role played by the emulsion polymerization of PVA is significant in the electrolytic polymerization of combined diene and non-diene rubbers with PVA as a surfactant. In addition, the surfactant of MF such as oleic acid has also role of the emulsion polymerization. Therefore, the possibility of using other surfactants in their electrolytic polymerization can be inferred. In general, the surfactants are categorized into four types as shown in Figure A2 (Appendix B): anionic, cationic, nonionic, and amphoteric. In this figure, the surfactants that can be used in electrolytic polymerization were designated a circle marks, whereas those that cannot were designated cross marks. The triangle mark denotes that the ability of electrolytic polymerization is different depending on the experimental condition. These results are shown in Table 1. A 312 mT magnetic field and an electric field at 20 V and 2.7 A were applied for 5 min with a 1 mm gap between electrodes. In this study, to investigate the possibility of mixing and electrolytic polymerization with NR-latex, the authors address Q rubber. Q can be divided into four types: KE1300T (Shin-Etsu Chemical Co. Ltd.), KE1400 (Shin-Etsu Chemical Co. Ltd.), KF96 (1000 cSt), and KF96 (1 cSt). KE1400 and KE1300T are silicone oils with some silane compounds and they are different from KF96, which is a pure silicone oil without any silane. Therefore, they are solidified to become solid silicone rubber using a curing agent. 

According to the molecular weight of PDMS, the viscosity of KF96 changes: KF96 of 1 cSt is smaller than KF96 of 1000 cSt. The authors used 3 g Q, 0.75 g surfactant, 0.75 g MF (W40), 3 g NR-latex, and 3 g Ni. The mixing type used was Q, surfactant, NR-latex, and Ni; and Q, surfactant, MF, NR-latex, and Ni. KE1400 and KE1300T, which are predominantly silicone oils with some silane, cannot be mixed with NR-latex. If NR-latex is not mixed, the rubber liquid cannot be electrolytically polymerized. Additionally, anionic surfactant contributes to the mixing of water-solvent and water-insolvent rubbers, including the electrolytic polymerization of the mixed rubber. The contributions made by the anionic surfactant is more than that made by the cationic one. These outcomes occur owing to the emulsion polymerization of the surfactant. Consequently, electrolytic polymerization is created by a surfactant; however, the kind of surfactant to be used is a significant factor to consider.

## 3. Consummate Fabrication of MCF Rubber Sensor

By foregoing the electrolytic polymerization of MCF rubber, whether it is diene or non-diene rubber, rubber can be solidified differently from conventional vulcanization methods using a surfactant. Next, the authors manage the problem of the consummate fabrication of the sensor by utilizing MCF rubber. This fabrication has two prerequisites of adhesion to electrodes and doping conditions.

Because sensing is performed by the measurement of electrical signals, electrodes need to be installed inside a sensor, enough to strictly adhere to the sensor. Regarding MCF rubber sensor, plate or wire type electrodes are required not to be removed from the rubber. Consequently, a novel adhesion technique of using a hydrate such as Na_2_WO_4_·2H_2_O is proposed [14]. The metal complex hydrate used is ionized in the compounded liquid with the MCF rubber, making the ion WO_4_^2–^ to become radical and vulcanized with isoprene molecules of water-soluble rubber by electrolytic polymerization. Additionally, on one hand, the MCF rubber adheres to the metal of the electrode at the anode side. On the other hand, to make the sensor exhibit varied sensing, force, temperature, and electromagnetic wave involving light doping or filling is quite often adopted. Mostly, the former used a chemical reagent as the dopant, whereas the latter used particle or powder as the filler. Regarding the MCF rubber sensor, various kinds of dopants were proposed and classified into three types: conductive, piezo, and battery types, as shown in Table A1 (Appendix C) [14]. The TiO_2_ or Ni involved in the MCF rubber corresponded to the filler.

However, the optimum adjustment of the conditions of the voltage and electric current of an electric field, its application period, magnetic field strength, electrodes gap, kind of electrodes, each component of the MCF rubber, and dopant on the electrolytic polymerization is needed so that the typical MCF rubber’s characteristics might be exhibited. This is also because of the aridity caused by extreme electrolytic polymerization and the impossibility of electrolytic polymerization, whose phenomena are initial solidification at the mixing, owing to non-congeniality between the MCF rubber and dopant. Therefore, to resolve this doping condition, the authors propose a new technique that involves doping directly into the solid rubber.

### 3.1. Rubber Stocking

Because solid rubber is generally impermeable, conventional solid rubber cannot be directly doped by any reagent. However, any liquid can permeate the electrolytically polymerized MCF rubber, as shown in Figure 16. KF96 contains 1000 cSt, whereas MF is W40. A 312 mT magnetic field and an electric field at 30 V and 2.7 A are applied for 5 min with a 1 mm gap between electrodes. These figures show water-soluble types and mixtures of water-soluble and water-insoluble rubber types with MCF rubbers. In Figure 16b,d, water can be seen as the shimmering reflection of light on the rubber surface after permeation.

However, as lapsing occurs after a long time of electrolytic polymerization, the MCF rubber is arid such that it takes a long time for water to permeate into the MCF rubber. Subsequently, when the MCF rubber is immersed in some liquid and the ambiance is evacuated, the liquid can be permeated into the MCF rubber. Furthermore, if the MCF rubber gets much more porous, more liquid can be permeated into the MCF rubber. Whether the MCF rubber is a water-soluble type or a combination of water-soluble and water-insoluble rubber type, by combining either metal complex hydrate Na_2_WO_4_·2H_2_O with water, or mixing their blend, MCF rubber got much more porous such as in stocking. For example, many stocking- or sponge-like porosis can be seen, as shown in image of Figure 17ao which is magnified on Figure 17k by scanning electron microscope (SEM). Compared to the MCF rubber without porosity, as shown in Figure 17a–e corresponding to Figure 16a,b, the MCF rubber stocking is shown in Figure 17f–an. A 312 mT magnetic field and an electric field at 30 V and 2.7 A are applied for 5 min with a 1 mm gap between the electrodes. By adding metal complex hydrate Na_2_WO_4_·2H_2_O, porosity gets proliferated but by adding water, they multiply, as seen from the image of rubber transmitted by the light in Figure 17a,f,k,p,u,z,ae,aj. The porosity is so large that the authors can confirm its existence in the rubber. Multiple repeated electrolytic polymerizations mean that the MCF rubber liquid has the same ingredient smothered on the once electrolytically polymerized MCF rubber, and then electrolytically polymerized again, such that the electrolytically polymerized MCF rubber creates multi-layers. Therefore, by comparing Figure 17k,p,u or Figure 17z,ae,aj, it can be deduced that the more electrolytic polymerization conducted, the less porosity created. The MCF rubber with many porous spaces can contain liquids by evacuation. Examples of microscopic images of the MCF rubber permeated by a reagent is presented in Figure 18. The MCF rubber was electrolytically polymerized thrice, where KI + I_2_ represents the KI solution, which is an aqueous potassium iodide solution made up of 60 g of water and 40 g of potassium iodide. KF96 contains 1000 cSt, whereas the MF used is W40. A 312 mT magnetic field and an electric field at 30 V and 2.7 A, are applied for 5 min with a 1 mm gap between the electrodes.

In the production of MCF rubber stocking, many peculiarities can be confirmed: the temperature and electrical properties during the electrolytic polymerization, and its adhesion to the electrodes as follows. Firstly, Figure 19 shows the time-lapse from the start of the application of an electric field, which indicates “on,” where MCF rubber 1 is represented in the figure corresponding to Figure 17a–e (0.75 g MF, 0.5 g TiO_2_, 3 g S-500, 3 g 671A, and 3 g Ni), MCF rubber 2 corresponding to Figure 17f–j (0.5 g Na_2_WO_4_·2H_2_O, MF, 0.5 g TiO_2_, 3 g S-500, 3 g 671A, and 3 g Ni), and MCF rubber 3 corresponding to Figure 17k–y (0.5 g Na_2_WO_4_·2H_2_O, 3 g water, 0.75 g MF, 0.5 g TiO_2_, 3 g S-500, 3 g 671A, and 3 g Ni). MF is W40. A 312 mT magnetic field and an electric field at 30 V and 2.7 A are applied for 5 min with a 1 mm gap between electrodes. By adding metal complex hydrate Na_2_WO_4_·2H_2_O and more water, temperature becomes higher and it takes a longer time for electric current to culminate and for voltage to reach 30 V. The enhancement of temperature and delay occurs because the process of electrolytic polymerization takes time.

Secondly, the electrolytically polymerized MCF rubber strictly adheres to the anode such that it is difficult to detach the MCF rubber from the electrode. This peculiarity differs according to the kind of electrode used. The strict adhesion occurs on stainless steel, aluminum, titanium, nickel, zinc, lead, brass, iron, and copper. This adhesion occurs owing to the balance between the work function and redox potentials of the metal of the electrode, and the metal complex hydrate Na_2_WO_4_·2H_2_O, which has been clarified in the previous study [14]. In contrast, the MCF rubber cannot adhere to titanium and chromium. Therefore, judging from the ease of detaching the MCF rubber from the electrode, these materials should be used. However, the possibility of solidifying the MCF rubber liquid by electrolytic polymerization varies depending on the constituents of the MCF rubber. Table 2 shows the results of solidification by electrolytic polymerization and the color of anode after electrolytic polymerization in the case titanium. In the case of chromium, the MCF rubber can be solidified by electrolytic polymerization regardless of any constituent of the MCF rubber. KF96 has 1000 cSt and MF is W40. A 312 mT magnetic field and an electric field at 20 V (or 30 V) and 2.7 A are applied for 5 min with a 1 mm gap between the electrodes. The liquid dye presented in the table is always used in a solar cell.

In the case of titanium, just anode electrode changes blue color purple, as shown in Table 2 and Figure 20. This phenomenon is related to the titanium oxide coating. Therefore, by utilizing MCF rubber, titanium can be coated. This can be proposed as another novel technique of utilizing MCF rubber.

Rubber stocking is useful in diverse ways except as the infiltrating material in the present MCF rubber sensing, for example, it is effective for filter and odor sensing [26]. As for the latter, the voltage between the ends of the rubber stocking, which is changed by the ionized molecular inserted into the hollows, can be measured. 

### 3.2. Consummate Fabrication

Using the foregoing percolation technique of permeating a reagent into the MCF rubber, as well as the adhesion technique of electrode wires on the MCF rubber of the previous study [14], a consummate fabrication of the MCF rubber sensor is proposed, as shown in Figure 21. For the adhesion technique produced “(a) adhesive MCF rubber 2” as shown in the figure preliminarily. By the percolation technique, porous MCF rubber 1 is electrolytically polymerized three times and permeated by a reagent: “(c) porous MCF rubber 1” as shown in the figure. However, the authors produce “(e) electrolytically polymerized MCF rubber 3” as shown in the figure, and this rubber does not have Na_2_WO_4_·2H_2_O and water because the roles of adhesion and infiltration used are not needed. “(e) electrolytically polymerized MCF rubber 3” is for the outer enclosure, and “(c) porous MCF rubber 1” is squeezed between them using liquid “(a) adhesive MCF rubber 2” and electrolytic polymerization, where electric wires are inserted inside the “(a) adhesive MCF rubber 2”. Next, to prevent the MCF rubber sensor from drying, it is coated with silicone oil rubber (KE1300T) with curing reagent. However, the surface of the produced MCF rubber sensor and the silicone oil rubber react to each other and cause them not to be solidified. This is another critical point of the production of the MCF rubber sensor. Therefore, “(d) liquid of MCF rubber 3” is initially coated on the produced MCF rubber sensor to be dried as “solidified MCF rubber 4” in the figure, and then the silicone oil rubber is coated to be dried.

One example of a specimen of the consummate MCF rubber sensor is shown in Figure 22a and its cross-section in Figure 22b. The MCF rubber inside the silicone oil rubber is 16 mm × 21.5 mm in size and 4 mm thick, and the size of the outer shell of silicone oil rubber is 16.5 mm × 24 mm and 4.5 mm thick.

As seen in previous sections, solid MCF rubber is permeable, which is a peculiar property. However, as regards to filling, to make it conductive, as shown in Table A1 (Appendix C), tin-plated thin and short copper wires (AWG12 type, Kyowa Harmonet Co. Ltd., Kyoto, Japan) with ϕ 0.12 mm diameter, and 2 mm length can be compounded in the MCF rubber by cutting them as fillers. Where the filler is viable to the inner compound of either “(a) adhesive MCF rubber 2” or “(c) porous rubber 1”. Figure 23 shows the induced voltage of the MCF rubber sensor to repeated pressure by the production of Figure 21 in the case of porous MCF rubber 1 without thin wires and the adhesive MCF rubber 2 case with thin wires. Varied reagents are percolated into the porous MCF rubber 1 by evacuation while varied particles are compounded in the porous MCF rubber 1 by mixing. KF 96 has 1000 cSt and MF is W40. A 312 mT magnetic field and an electric field at 2.7 A are applied with a 1 mm gap between the electrodes. The liquid of “(a) adhesive MCF rubber 2” contains 0.5 g Na_2_WO_4_·2H_2_O, 3 g water, 0.75 g MF, 3 g NR-latex (Ulacol), 3 g CR-latex (671A), and 3 g Ni. As for liquid of “(b) porous MCF rubber 1”, it contains 0.5 g Na_2_WO_4_·2H_2_O, 3 g water, 0.75 g MF, 3 g PDMS (KF96), 3 g PVA, 3 g NR-latex (Ulacol), 3 g CR-latex (671A), 3 g TiO_2_, and 3 g Ni. In the case of the liquid for “(d) MCF rubber 3,” it contains 0.75 g MF, 3 g NR-latex (Ulacol), 3 g CR-latex (671A), 3 g TiO_2_, and 3 g Ni. BaTiO_3_ and Al_2_O_3_ (CR, 3 μm) are measured and used at 1 g.

The MCF rubber sensor is sensitive enough to be practical by the percolation technique with reagents into the MCF rubber. The trends show that the induced voltage involving dielectric particles such as Al_2_O_3_ and BaTiO_3_ and non-conducting liquid such as glycerin become the largest among others. BaTiO_3_ is often used in the piezo-element. The cases with sodium hexadecyl sulfate solution (C_16_H_33_NaO_4_S) which is anionic surfactant, with lithium hydroxide monohydrate (LiOH·2H_2_O) which is used as electrolyte of battery and lithium citrate tetrahydrate (Li_3_C_6_H_5_O_7_·4H_2_O), are the second largest. Glycerin (C_3_H_5_(OH)_3_) is one emulsifier. Then the induced voltage with surfactant or emulsifier tends to be large. On the other hand, the induced voltage with particle-type dopant tends to be larger than the one with liquid-type. The other cases involving alkali aqueous solution such as KOH and KI+I_2_, water, kerosene or kerosene-type liquid such as MSGS60, are the smallest.

## 4. Conclusions

The present first report offers the key to an understanding of practicability of electrolytic polymerization for the solidification of rubber with various combinations of diverse kinds of rubber, MF and emulsifier. And another critical point is about the fabrication of the MCF rubber sensor with their various combinations and the porous rubber permeated by any dopants. The production of the porous rubber (rubber stocking) and the permeation of any liquids into the rubber are also novel technique of the present study.

The authors presented the role of the emulsion polymerization of PVA, which is significant in the electrolytic polymerization of combined diene and non-diene rubbers. By dealing with U as another non-diene rubber except for Q, the electrolytic polymerization technique is feasible, in which PVA is utilized and water-soluble diene rubber (NR-latex or CR-latex) is mixed with non-diene rubber. The Diene type MCF rubber mixed by water-insoluble liquid or water-insoluble MF using the emulsion polymerization of PVA had the feasibility of haptic sensing; therefore, non-diene type MCF rubber combined by mixing water-insoluble or water-based MF, with soluble rubber such as NR-latex or CR-latex, using the emulsion polymerization of PVA, can have the feasibility of sensing. Consequently, under the diverse combination of soluble and insoluble rubber, liquid or MF too, electrolytic polymerization can be conducted on soluble and insoluble rubbers, with insoluble rubber and MF, or soluble and insoluble MFs in the rubber. Additionally, the MCF rubber sensor produced by those compounded materials has a feasibility of sensing pressure as piezo-electricity and piezo-resistivity.

The utilization of PVA enables what was considered impossible until now, allowing the water-soluble rubber to combine with water-insoluble rubber or liquid to form a practical mixture. Because PVA is a surfactant, other surfactants are feasible in their electrolytic polymerization. In particular, the anionic surfactant tends to contribute to the mixing of water-soluble and water-insoluble rubbers, and the electrolytic polymerization of the mixed rubber.

Conventional solid rubber cannot be directly doped by any reagent because solid rubber is generally impermeable. However, any liquid can permeate the electrolytically polymerized MCF rubber. Whether the MCF rubber is a water-soluble type or a combination of water-soluble and water-insoluble rubber types, by combining either metal complex hydrate Na_2_WO_4_·2H_2_O or water, or by combining their mixture, MCF rubber becomes much more porous such as stocking. The rubber stocking, which can also be percolated by any liquid, has certain peculiarities: the enhancement of temperature and the long period of electrolytic polymerization, and the strict adhesion of the electrolytically polymerized MCF rubber to the anode, which detaches the MCF rubber from the electrode. The percolation technique of permeating a reagent into the MCF rubber stocking is useful in the case of the infiltrating material to rubber.

Using the rubber stocking and the previously proposed adhesion technique of electric wires to the MCF rubber, the authors constructed a consummate fabrication process of the MCF rubber sensor. The MCF rubber sensor is sensitive so that the percolation technique of permeating a reagent into the MCF rubber is productive. The diverse specified examples of using the MCF rubber sensor are presented in another sequential second report [24].

## Figures and Tables

**Figure 1 sensors-20-04658-f001:**
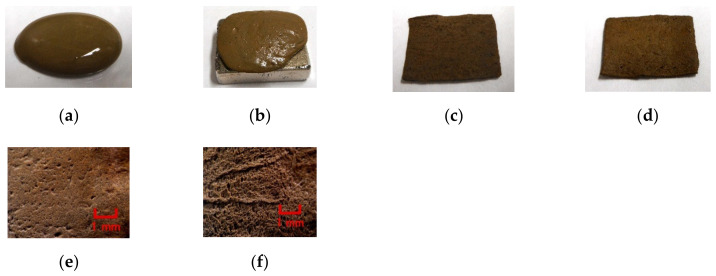
Images of liquid MCF rubber before electrolytic polymerization and electrolytically polymerized MCF rubber containing U: (**a**) liquid MCF rubber before electrolytic polymerization without a magnetic field; (**b**) liquid MCF rubber before electrolytic polymerization under a magnetic field; (**c**) panoramic image of electrolytically polymerized MCF rubber facing anode; (**d**) panoramic image of electrolytically polymerized MCF rubber facing cathode; (**e**) microscopic image of electrolytically polymerized MCF rubber facing anode by optical microscope; (**f**) microscopic image of electrolytically polymerized MCF rubber facing cathode by optical microscope.

**Figure 2 sensors-20-04658-f002:**
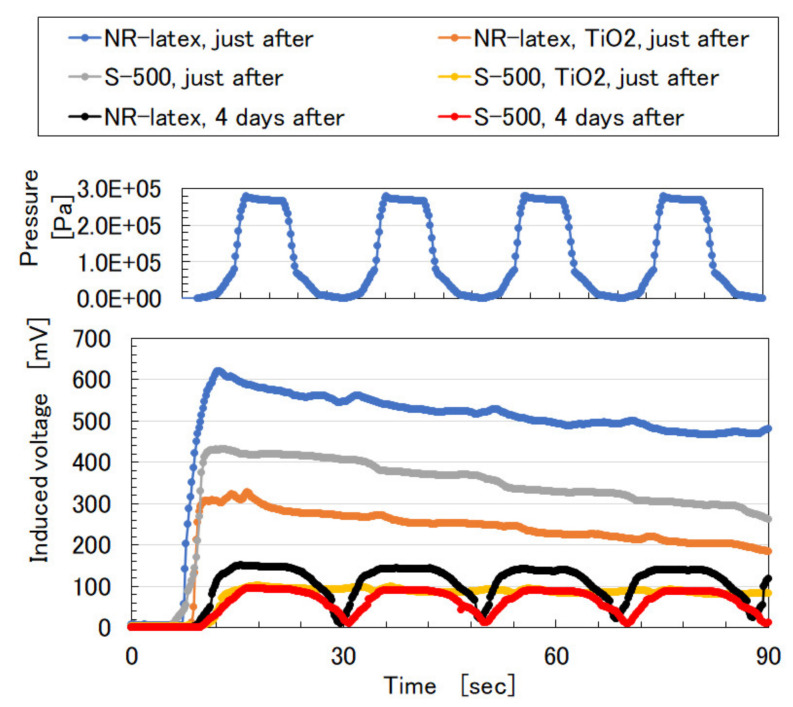
Induced voltage of MCF rubber with U to repeat pressure application: either NR-latex or S-500 is compounded; TiO_2_ presents the use of TiO_2_; “just after” means the measurement just after the production; “4 days after” means the measurement from the production after 4 days.

**Figure 3 sensors-20-04658-f003:**
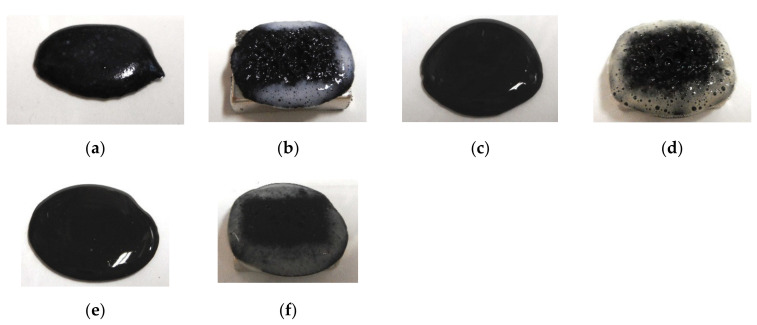
Images of liquid mixed with water-insoluble liquid and water: (**a**) kerosene mixed without a magnetic field; (**b**) kerosene mixed under a magnetic field; (**c**) alkyl naphthalene mixed without a magnetic field; (**d**) alkyl naphthalene mixed under a magnetic field; (**e**) paraffin mixed without a magnetic field; (**f**) paraffin mixed under a magnetic field.

**Figure 4 sensors-20-04658-f004:**
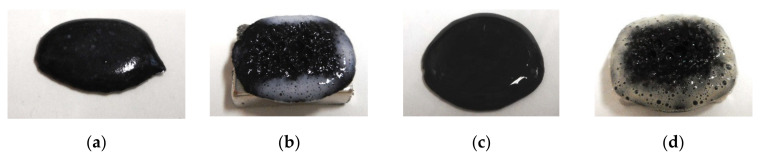
Images of liquid mixed water-insoluble MF and water-based MF: (**a**) kerosene-based MF mixed without a magnetic field; (**b**) kerosene-based MF mixed under a magnetic field; (**c**) alkyl naphthalene-based MF mixed without a magnetic field; (**d**) alkyl naphthalene-based MF mixed under a magnetic field.

**Figure 5 sensors-20-04658-f005:**
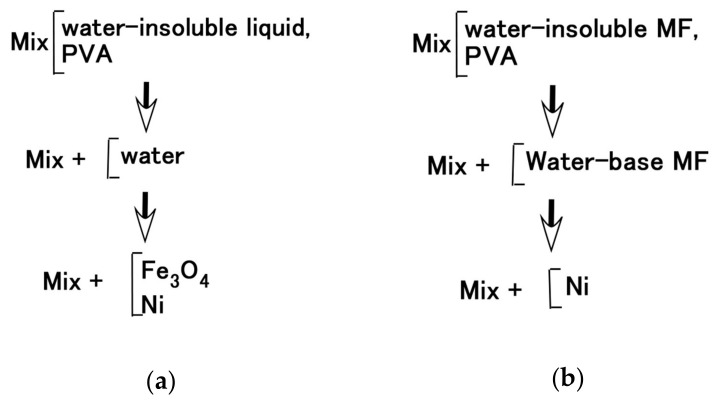
Mixing process; (**a**) for Figure 3; (**b**) for Figure 4.

**Figure 6 sensors-20-04658-f006:**
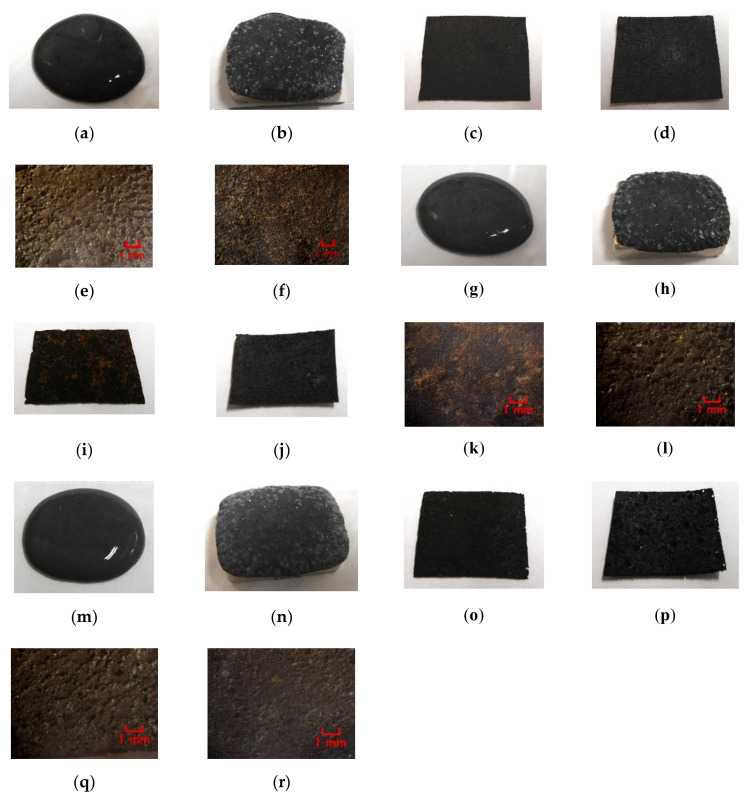
Images of liquid before electrolytic polymerization and mixing electrolytically polymerized rubber with water-insoluble liquid: (**a**–**f**) kerosene; (**g**–**l**) alkyl naphthalene; (**m**–**r**) paraffin; (**a**,**g**,**m**) liquid before electrolytic polymerization without magnetic field; (**b**,**h**,**n**) liquid before electrolytic polymerization under magnetic field; (**c**,**i**,**o**) panoramic image of electrolytically polymerized rubber facing anode; (**d**,**j**,**p**) panoramic image of electrolytically polymerized rubber facing cathode; (**e**,**k**,**q**) microscopic image of electrolytically polymerized rubber facing anode by optical microscope; (**f**,**l**,**r**) microscopic image of electrolytically polymerized rubber facing cathode by optical microscope.

**Figure 7 sensors-20-04658-f007:**
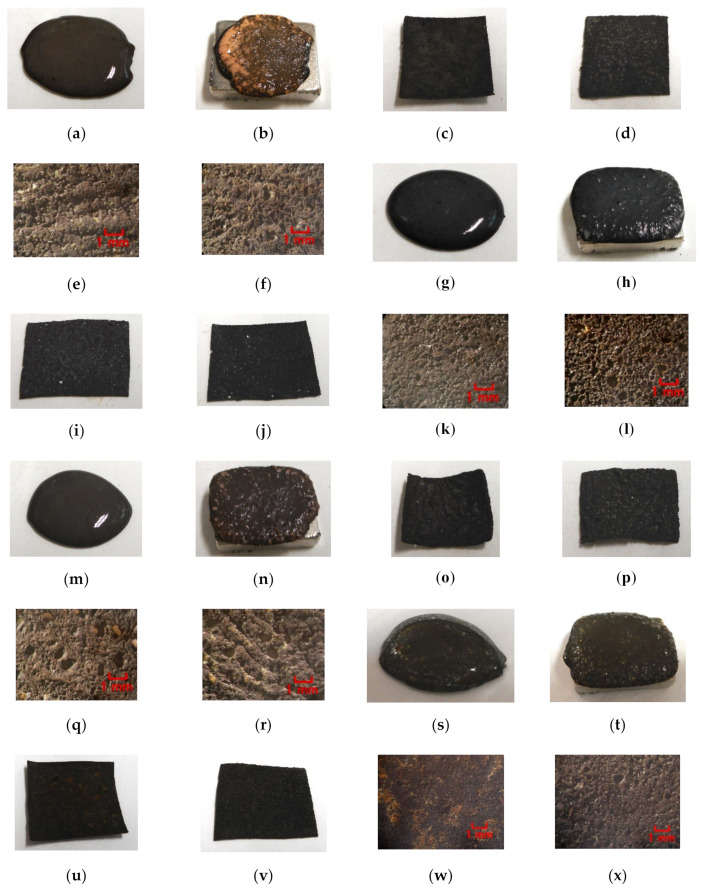
Images of liquid before electrolytic polymerization and mixing electrolytically polymerized rubber with water-insoluble MF: (**a**–**f**) MSGS60; (**g**–**l**) A500; (**m**–**r**) HC50; (**s**–**x**) DS50; (**a**,**g**,**m**,**s**) liquid before electrolytic polymerization without magnetic field; (**b**,**h**,**n**,**t**) liquid before electrolytic polymerization under magnetic field; (**c**,**i**,**o**,**u**) panoramic image of electrolytically polymerized rubber facing anode; (**d**,**j**,**p**,**v**) panoramic image of electrolytically polymerized rubber facing cathode; (**e**,**k**,**q**,**w**) microscopic image of electrolytically polymerized rubber facing anode by optical microscope; (**f**,**l**,**r**,**x**) microscopic image of electrolytically polymerized rubber facing cathode by optical microscope.

**Figure 8 sensors-20-04658-f008:**
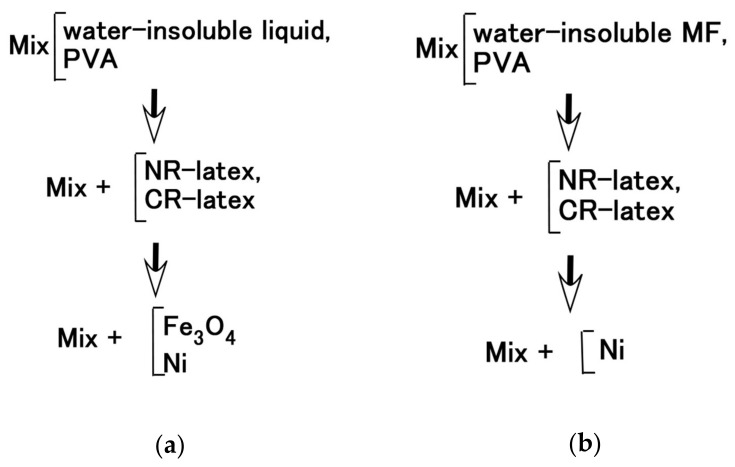
Mixing process for: (**a**) Figure 6; (**b**) Figure 7.

**Figure 9 sensors-20-04658-f009:**
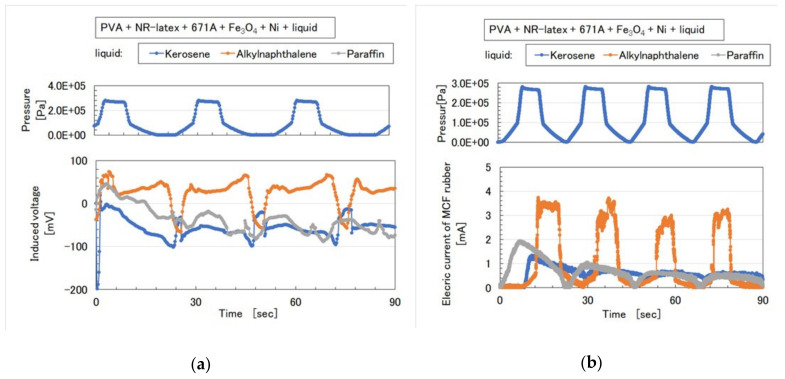
Induced voltage and electric current as piezo-resistivity of MCF rubber with water-insoluble liquid to repeat pressure application in Figure 6: (**a**) piezo-electricity; (**b**) piezo-resistivity.

**Figure 10 sensors-20-04658-f010:**
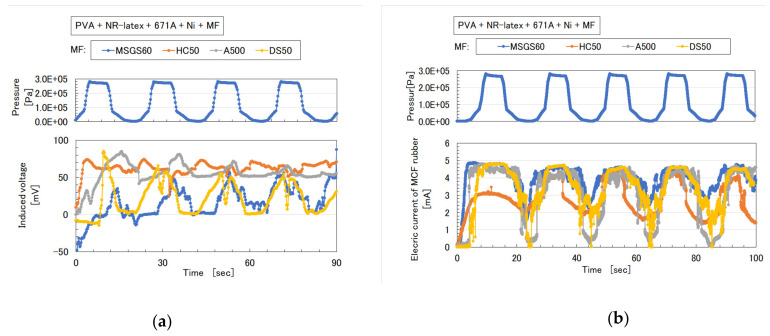
Induced voltage and electric current as piezo-resistivity of MCF rubber with water-insoluble MF to repeat pressure application in Figure 7: (**a**) piezo-electricity; (**b**) piezo-resistivity.

**Figure 11 sensors-20-04658-f011:**
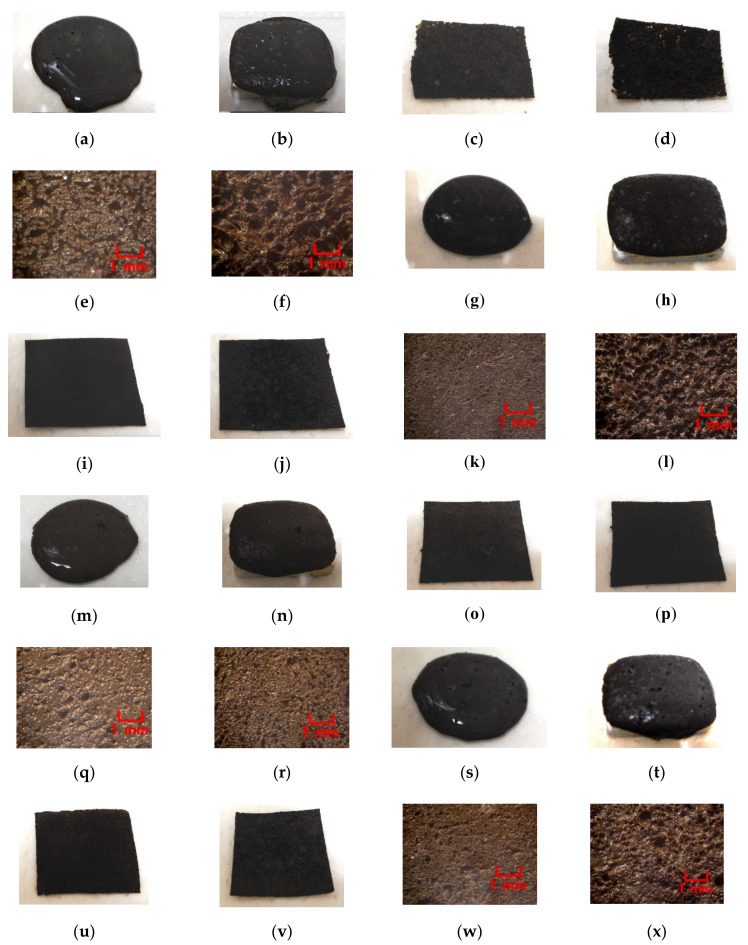
Images of liquid before electrolytic polymerization and electrolytically polymerized rubber mixed with KF96 and water-insoluble MF: (**a**–**f**) MSGS60; (**g**–**l**) A500; (**m**–**r**) HC50; (**s**–**x**) DS50; (**a**,**g**,**m**,**s**) liquid before electrolytic polymerization without magnetic field; (**b**,**h**,**n**,**t**) liquid before electrolytic polymerization under magnetic field; (**c**,**i**,**o**,**u**) panoramic image of electrolytically polymerized rubber facing anode; (**d**,**j**,**p**,**v**) panoramic image of electrolytically polymerized rubber facing cathode; (**e**,**k**,**q**,**w**) microscopic image of electrolytically polymerized rubber facing anode by optical microscope; (**f**,**l**,**r**,**x**) microscopic image of electrolytically polymerized rubber facing cathode by optical microscope.

**Figure 12 sensors-20-04658-f012:**
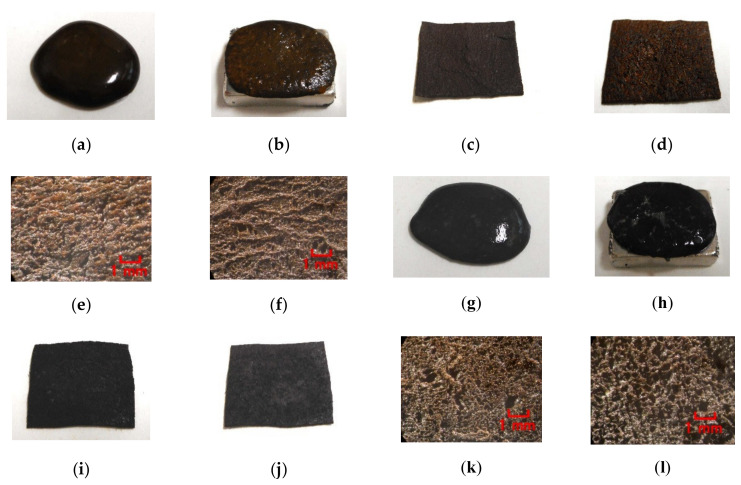
Images of liquid before electrolytic polymerization and electrolytically polymerized rubber mixed with KF96, W40, water-insoluble MF: (**a**–**f**) MSGS60; (**g**–**l**) A500; (**a**,**g**) liquid before electrolytic polymerization without magnetic field; (**b**,**h**) liquid before electrolytic polymerization under magnetic field; (**c**,**i**) panoramic image of electrolytically polymerized rubber facing anode; (**d**,**j**) panoramic image of electrolytically polymerized rubber facing cathode; (**e**,**k**) microscopic image of electrolytically polymerized rubber facing anode by optical microscope; (**f**,**l**) microscopic image of electrolytically polymerized rubber facing cathode by optical microscope.

**Figure 13 sensors-20-04658-f013:**
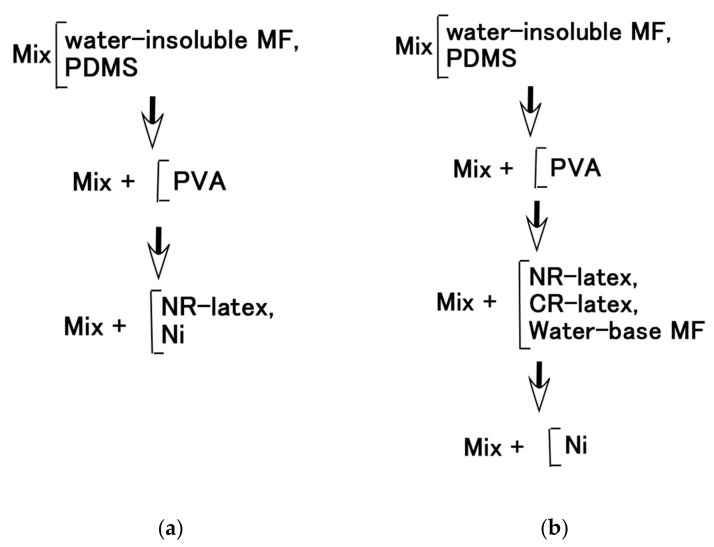
Mixing process for: (**a**) Figure 11; (**b**) Figure 12.

**Figure 14 sensors-20-04658-f014:**
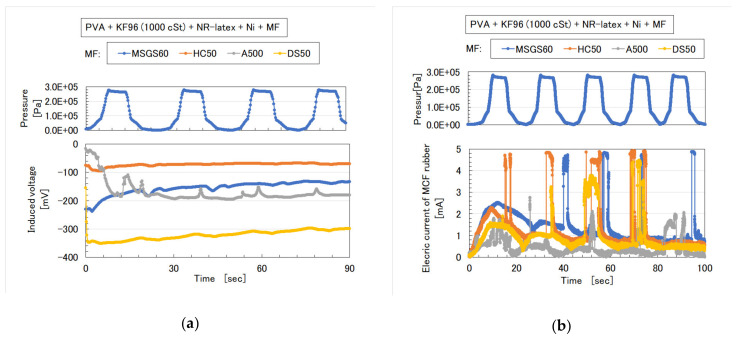
Induced voltage and electric current as piezo-resistivity of MCF rubber with KF96 and water-insoluble MF to repeat application of pressure in Figure 6: (**a**) piezo-electricity; (**b**) piezo-resistivity.

**Figure 15 sensors-20-04658-f015:**
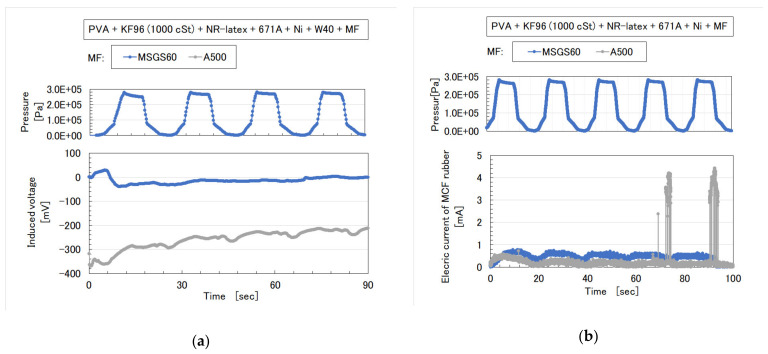
Induced voltage and electric current as piezo-resistivity of MCF rubber with KF96, water-insoluble MF, and water-based MF to repeat application of pressure in Figure 7: (**a**) piezo-electricity; (**b**) piezo-resistivity.

**Figure 16 sensors-20-04658-f016:**
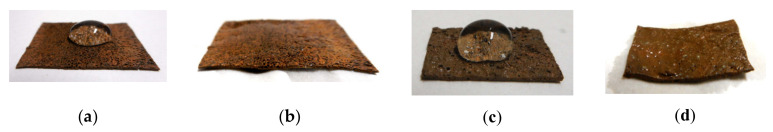
Images of MCF rubber before and after water permeation: (**a**) before permeation (0.75 g MF, 0.5 g TiO_2_, 3 g S-500, 3 g 671A, and 3 g Ni); (**b**) after permeation (0.75 g MF, 0.5 g TiO_2_, 3 g S-500, 3 g 671A, and 3 g Ni); (**c**) before permeation (3 g KF96, 3 g PVA, 0.75 g MF, 0.5 g TiO_2_, 3 g S-500, 3 g 671A, and 3 g Ni); (**d**) after permeation (3 g KF96, 3 g PVA, 0.75 g MF, 0.5 g TiO_2_, 3 g S-500, 3 g 671A, and 3 g Ni).

**Figure 17 sensors-20-04658-f017:**
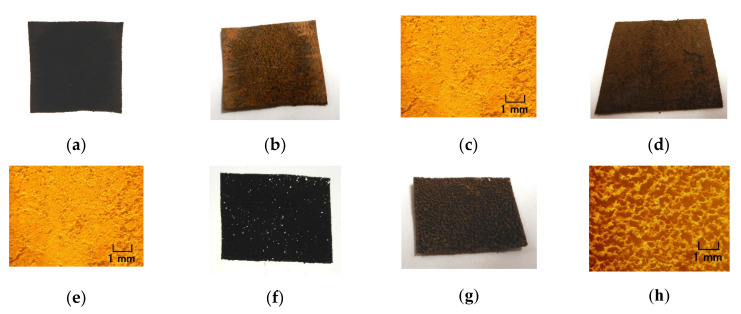
Images of electrolytically polymerized rubber: (**a**–**e**) 0.75 g MF, 0.5 g TiO_2_, 3 g S-500, 3 g 671A, and 3 g Ni; (**f**–**j**) 0.5 g Na_2_WO_4_·2H_2_O, 0.75 g MF, 0.5 g TiO_2_, 3 g S-500, 3 g 671A, and 3 g Ni; (**k**–**y**) 0.5 g Na_2_WO_4_·2H_2_O, 3 g water, 0.75 g MF, 0.5 g TiO_2_, 3 g S-500, 3 g 671A, and 3 g Ni; (**z**–**an**) 0.5 g Na_2_WO_4_·2H_2_O, 3 g water, 3 g KF96 (1000 cSt), 3 g PVA, 0.75 g MF, 0.5 g TiO_2_, 3 g S-500, 3 g 671A, and 3 g Ni; (**a**–**o**), (**z**–**ad**) electrolytic polymerization conducted one time; (**p**–**t**,**ae**–**ai**) electrolytic polymerization conducted two times; (**u**–**y**,**aj**–**an**) electrolytic polymerization conducted three times; (**a**,**f**,**k**,**p**,**u**,**z**,**ae**,**aj**) panoramic image of rubber facing cathode transmitted by light; (**b**,**g**,**l**,**q**,**v**,**aa**,**af**,**ak**) panoramic image of rubber facing cathode; (**c**,**h**,**m**,**r**,**w**,**ab**,**ag**,**al**) microscopic image of rubber facing cathode by optical microscope; (**d**,**i**,**n**,**s**,**x**,**ac**,**ah**,**am**) panoramic image of rubber facing anode; (**e**,**j**,**o**,**t**,**y**,**ad**,**ai**,**an**) microscopic image of rubber facing anode by optical microscope; (**ao**) microscopic image of (**k**) by SEM with 50 × magnification.

**Figure 18 sensors-20-04658-f018:**
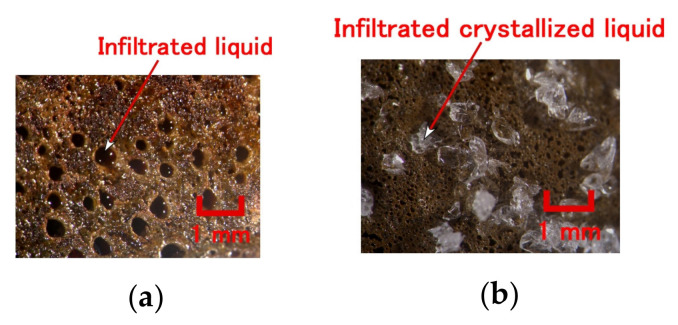
Images of porous MCF rubber after percolation by evacuation (0.5 g Na_2_WO_4_·2H_2_O, 3 g KF96, 3 g PVA, 0.75 g MF, 0.5 g TiO_2_, 3 g S-500, 3 g 671A, and 3 g Ni): (**a**) KI+I_2_; (**b**) adipic acid.

**Figure 19 sensors-20-04658-f019:**
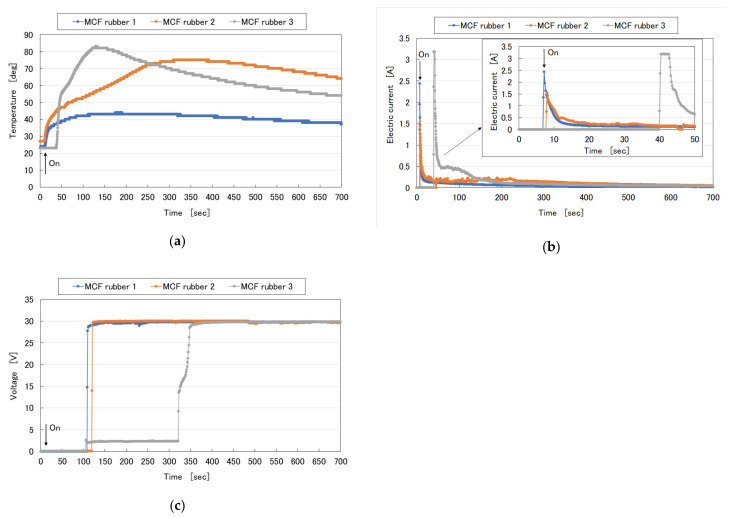
Change in temperature, electric current, and voltage of MCF rubber liquid during electrolytic polymerization: (**a**) temperature; (**b**) current; (**c**) voltage.

**Figure 20 sensors-20-04658-f020:**
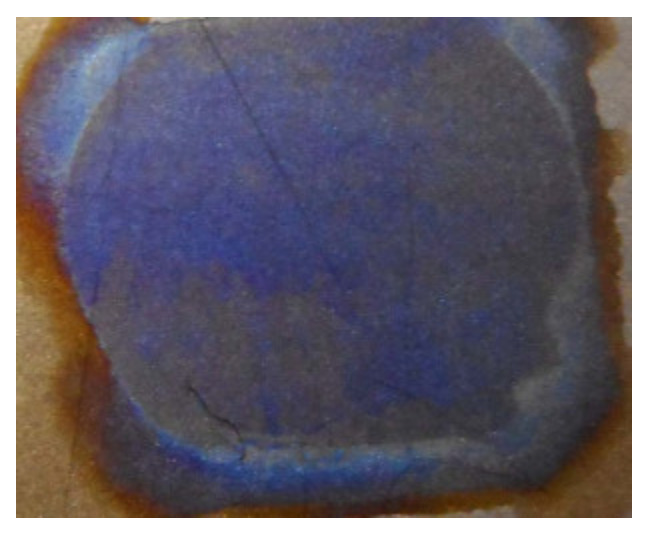
Images of titanium anode after electrolytic polymerization designating 9 in Table 2.

**Figure 21 sensors-20-04658-f021:**
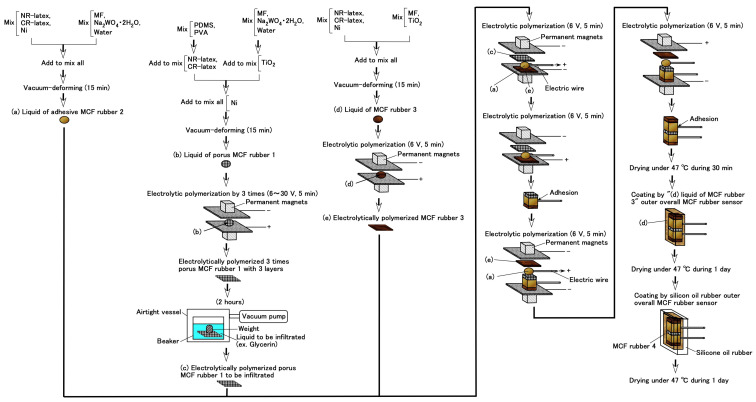
Production procedure of consummate fabrication of MCF rubber sensor.

**Figure 22 sensors-20-04658-f022:**
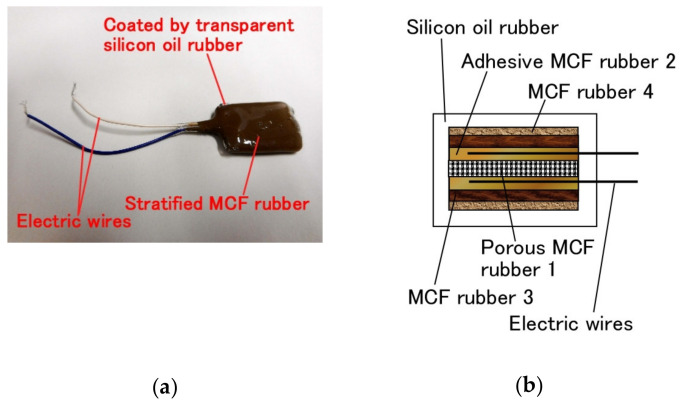
Specimen of the consummate MCF rubber sensor by the production process of Figure 21: (**a**) photograph; (**b**) schematic cross section.

**Figure 23 sensors-20-04658-f023:**
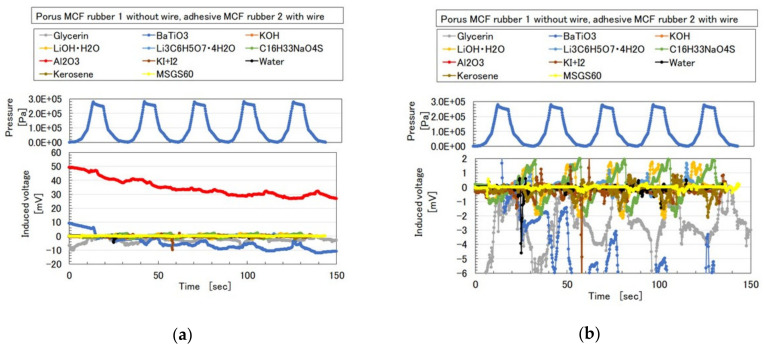
Induced voltage of MCF rubber sensor to repeated pressure by the production process of Figure 21: (**a**) zooming out of (**b**); (**b**) zooming in from (**a**).

**Table 1 sensors-20-04658-t001:** Results of the possibility of mixing and electrolytic polymerization with surfactant and Q rubber liquid.

Surfactant	Two types of Mixing	Mixing + KE1300T	Mixing + KE1400	Mixing + KF96 (1000 cSt)	Mixing + KF96 (1 cSt)
Sodium oleate solution ^3^	+ NR-latex	✕ ^1^	✕	◯ ^2^	◯
	+ NR-latex, MF	✕	✕	◯	◯
Sodium lauric solution ^3^	+ NR-latex	✕	✕	◯	◯
	+ NR-latex, MF	✕	✕	◯	◯
Sodium naphthalene sulfonate solution ^3^	+ NR-latex	✕	✕	✕	◯
	+ NR-latex, MF	✕	✕	✕	✕
Sodium hexadesyl sulfate solution ^3^	+ NR-latex	✕	✕	◯	◯
	+ NR-latex, MF	✕	✕	◯	✕
Sodium dodesyl sulfate solution ^3^	+ NR-latex	✕	✕	◯	◯
	+ NR-latex, MF	✕	✕	◯	◯
Tetramethylammonium hydroxy solution ^4^	+ NR-latex	✕	✕	✕	✕
	+ NR-latex, MF	◯	✕	◯	✕
Benzethonium chloride ^4^	+ NR-latex	✕	✕	◯	✕
	+ NR-latex, MF	✕	✕	✕	✕
Methylammonium chloride solution ^4^	+ NR-latex	✕	✕	✕	✕
	+ NR-latex, MF	✕	✕	◯	✕
Glycerol monostearic acid solution ^5^	+ NR-latex	◯	✕	◯	◯
	+ NR-latex, MF	✕	✕	◯	
Lauryl dimethylamylacetic acid solution ^6^	+ NR-latex	✕	✕	◯	◯
	+ NR-latex, MF	✕	✕	◯	◯
N,N-dimethyldesylamine N-oxide solution ^6^	+ NR-latex	✕	✕	◯	◯
	+ NR-latex, MF	✕	✕	◯	◯

^1^ Q rubber liquid cannot be mixed beforehand; ^2^ Q rubber liquid can be mixed and solidified by electrolytic polymerization.; ^3^ anionic surfactant; ^4^ cationic surfactant; ^5^ nonionic surfactant; ^6^ amphoteric surfactant.

**Table 2 sensors-20-04658-t002:** Results of solidification by electrolytic polymerization and color of anode after the electrolytic polymerization in the case of titanium anode.

NR-Latex [g]	671A [g]	KF96 [g]	PVA [g]	MF [g]	Ni [g]	TiO_2_ [g]	Voltage [V]	Dopant [g]	Na_2_WO_4_·2H_2_O [g]	Rubber ^1^	Color ^2^
3				0.75	3		6			S ^4^	NC ^7^
3				0.75	3	0.5	6			S ^4^	NC ^7^
3	3	3	3	0.75	3		6			N ^5^	NC ^7^
3	3	3	3	0.75	3		20.30			PS ^6^	P ^8^
3	3	3	3	0.75	3		30	Dye ^3^		S ^4^	P ^8^
3	3	3	3	0.75	3	0.5	6			N ^5^	NC ^7^
3	3	3	3	0.75	3	0.5	20			N ^5^	P ^8, 9^
3	3	3	3	0.75	3	0.5	30			PS ^6^	P ^8^
3	3	3	3	0.75	3	0.5	30	Dye ^3^		S ^4^	P ^8^
3	3	3	3	0.75	3		30	Dye ^3^		S ^4^	P ^8^
3	3	3	3	0.75	3		6		0.5	N ^5^	NC ^7^
3	3	3	3	0.75	3		30		0.5	PS ^6^	P ^8^
3	3	3	3	0.75	3	0.5	6		0.5	N ^5^	NC ^7^
3	3	3	3	0.75	3	0.5	30		0.5	PS ^6^	P ^8^

^1^ electrolytic polymerization; ^2^ Color of anode after electrolytic polymerization; ^3^ Liquid dye based on Ruthenium complexes PEC-TOM-P04 (Peccell Technologies Co. Ltd., Yokohama, Japan); ^4^ S means solidified; ^5^ N means not solidified; ^6^ PS means partially solidified; ^7^ NC means not changing color; ^8^ P means changing to blue purple; ^9^ see Figure 20.

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
