# Peer review of "Enhancement of Diversity in Production and Application Utilizing Electrolytically Polymerized Rubber Sensors with MCF: 1st Report on Consummate Fabrication Combining Varied Kinds of Constituents with Porous Permeant Stocking-Like Rubber"

_sensors, 2020, doi:10.3390/s20174658_

Round 1
Reviewer 1 Report
Question 1: The paper under review presents very interesting experimental results concerning Enhancement of Diversity on Production and Application Utilizing Electrolytically Polymerized Rubber Sensors with MCF. But the paper is hard to read. It is suggested to have the paper edited by someone with expertise in technical English editing, paying particular attention to English grammar and sentence structuring.
Question 2: Please emphasize the novelty of the current work. Also, more information should be given about previous work in the similar area.
Question3: I would recommend a detailed discussion on Figure 23.
Author Response
Thank you for your valuable comments and suggestions for the present our report. According to their comments, we would like to reply for them as follows.
Question 1: According to your indication, we confirmed to revise English by technical English editing in the overall document. Incidentally, they are numerous and minute enough not to be highlighted in the revised manuscript.
Question 2: According to your indication, the explanations of the novelty of the current work and of the information about previous works by comparing to our study are added in the Introduction, Conclusion, and so on, which is for the effort to expand the range of studies considered and more clearly understood what place the current work occupies and which is the reply to the same another reviewing.
Question 3: According to your indication, we added the detail discussion and explanation on Fig. 23.

Reviewer 2 Report
Review of «Enhancement of Diversity on Production and Application Utilizing Electrolytically Polymerized Rubber Sensors…» by K. Shimada et al.
The article is devoted to the actual problem of development of composite rubber materials that can be used as an artificial skin for robotic technologies. Research is well conducted, of broad interest, and has many technical applications.
I have only one comment that should be addressed before publishing. The overview of the current state of the problem presented in the introduction is limited. The authors refer mainly to their own works. It is necessary to expand the range of studies considered and more clearly demonstrate what place the current work occupies in existing research on this topic.
Author Response
Thank you for your valuable comments and suggestions for the present our report. According to their comments, we would like to reply for them as follows.
According to your indication, the explanations of the novelty of the current work and of the information about previous works by comparing to our study are added in the Introduction, Conclusion, and so on, which is for the effort to expand the range of studies considered and more clearly understood what place the current work occupies and which is the reply to the same another reviewing.
